**Data Availability Statement:** The datasets supporting the conclusions of this article are available in the NCBI Sequence Read Archive BioProject PRJNA742396.

# Phenotypic and transcriptomic responses of cultivated sunflower seedlings (*Helianthus annuus* L.) to four abiotic stresses

**Max H. Barnhart**[ID]⍟*, **Rishi R. Masalia**⍟, **Liana J. Mosley**, **John M. Burke**[ID]

Department of Plant Biology, University of Georgia, Athens, GA, United States of America

⍟ These authors contributed equally to this work.
* maxbarnhart@uga.edu

## Abstract

Plants encounter and respond to numerous abiotic stresses during their lifetimes. These stresses are often related and could therefore elicit related responses. There are, however, relatively few detailed comparisons between multiple different stresses at the molecular level. Here, we investigated the phenotypic and transcriptomic response of cultivated sunflower (*Helianthus annuus* L.) seedlings to three water-related stresses (i.e., dry-down, an osmotic challenge, and salt stress), as well as a generalized low-nutrient stress. All four stresses negatively impacted seedling growth, with the nutrient stress having a more divergent response from control as compared to the water-related stresses. Phenotypic responses were consistent with expectations for growth in low-resource environments, including increased (i.e., less negative) carbon fractionation values and leaf C:N ratios, as well as increased belowground biomass allocation. The number of differentially expressed genes (DEGs) under stress was greater in leaf tissue, but roots exhibited a higher proportion of DEGs unique to individual stresses. Overall, the three water-related stresses had a more similar transcriptomic response to each other vs. nutrient stress, though this pattern was more pronounced in root vs. leaf tissue. In contrast to our DEG analyses, co-expression network analysis revealed that there was little indication of a shared response between the four stresses in despite the majority of DEGs being shared between multiple stresses. Importantly, osmotic stress, which is often used to simulate drought stress in experimental settings, had little transcriptomic resemblance to true water limitation (i.e., dry-down) in our study, calling into question its utility as a means for simulating drought.

## Background

Crop plants encounter a variety of abiotic stresses throughout their lives [e.g., 1–4]. Of these, water and nutrient limitation are amongst the most important, with both having large impacts on stand establishment and productivity [e.g., 2,5,6]. In the coming decades, climate change is expected to produce increasingly unpredictable precipitation patterns, resulting in longer and more frequent droughts [e.g., 4,7–10]. Agricultural zones are also expected to shift due to

**Funding:** This work was supported by a grant from the NSF Plant Genome Research Program (IOS-1444522; https://beta.nsf.gov/funding/opportunities/plant-genome-research-program-pgrp) to JMB. The funders had no role in study design, data collection and analysis, decision to publish, or preparation of the manuscript.

**Competing interests:** The authors have declared that no competing interests exist.

**Abbreviations:** PEG, polyethylene glycol 6000; DEGs, differentially expressed genes; LMF, leaf mass fraction; RMF, root mass fraction; SMF, shoot mass fraction; PCA, principal component analysis; LMA, leaf mass per area; RWC, relative water content; MDS, multidimensional scaling; GO, gene ontology.

urbanization and soil degradation [e.g., 11–14], placing additional strain on agricultural systems. While irrigation practices can be used to offset low precipitation, there are limits to water availability, and large-scale irrigation can result in soil salinization [3,15]. Similarly, though fertilizers can offset nutrient deprivation, they are economically, environmentally, and energetically costly [12,16–18] and in limited supply [16,19]). When coupled with increased agricultural demands due to a growing human population, water and nutrient limitations represent major challenges for long-term food security [3,20,21]. Understanding how plants respond to such stresses is thus a topic of great interest.

It is well established that, when exposed to low-resource environments, plants exhibit a suite of resource-conservative physiological and morphological responses, many of which result in slow growth [e.g., 22–25]. For example, plants in water-limited environments often decrease their stomatal conductance to conserve water, which could negatively impact growth through a reduction in available carbon [e.g., 26–28], while growth in nitrogen-depleted soils can limit the production of photosynthetic enzymes [22,23]. Plants grown in low-resource environments can respond by allocating more to organs involved in acquiring a limiting resource [23]. These phenotypic responses can be passive (i.e. traits that scale with mass) or active (i.e., traits that scale independently from mass; [29]). Importantly, this means that the effects of stress are often best observed using size-independent phenotypes which do not vary as a result of allometric scaling and differences in overall vigor [30–33]. In the case of water or nutrient limitation, increased root growth and altered root morphology (e.g., more root tips) can improve resource acquisition [e.g., 23,24]. Moreover, water and nutrient limitation can interact, as low water availability can limit nutrient uptake [e.g., 34–36]; conversely, root hydraulic conductance can be influenced by nutrient limitation [37]. Despite the occurrence of common phenotypic responses to stress, there are many examples of stress-specific responses. For instance, drought and salt stress both reduce osmotic potential, while salt can have harmful effects due to the uptake of toxic inorganic ions [4,38–40].

At the molecular level, one of the most fundamental ways in which plants respond to environmental challenges is to modulate gene expression [e.g., 41–46]. As such, researchers have often focused on the identification of genes that are differentially expressed in response to stress [e.g., 47–53]. In this context, differentially expressed genes (DEGs) can be characterized as being stress-specific, shared across multiple stresses, or induced by a particular combination of stresses [e.g., 54–56]. It has been argued that genes shared across multiple stress scenarios are of particular interest because of their potential role in the response to multiple disparate stresses, perhaps due to their involvement in common stress-signaling pathways [57–60]. While these sorts of analyses were originally conducted using microarrays [e.g., 43,52,59,61–68], such work now relies on RNA-sequencing as an unbiased means for investigating the transcriptional response to stress [e.g., 47,49,53,69–71]. In general terms, such studies have revealed that there are significant changes to gene expression levels under stress and that these changes can be quite different between tissue types [72–78]. Although many studies have investigated the transcriptomic response to a single stress in multiple tissues, fewer have compared the transcriptomic response across multiple different abiotic stresses in isolation (but see, [e.g., 79–83] for experimental examples and [84–87]).

Here, we describe a series of analyses aimed at characterizing the phenotypic and transcriptomic responses of cultivated sunflower seedlings (*Helianthus annuus* L.) to multiple different stress scenarios. Cultivated sunflower is an important oilseed crop that is often grown in rainfed regions [88]. Water and nutrient limitation at the seedling stage can severely limit stand establishment in sunflower, thereby greatly reducing yields [88,89]. The focal stresses in this study included three water-related stresses: a repeated dry-down, an osmotic stress implemented using polyethylene glycol 6000 (hereby referred to as PEG–this is a commonly used

agent for simulating drought stress; [e.g., 90–94], and salt [NaCl] stress. We also included a generalized low-nutrient stress to compare against the three water-related stresses. We measured numerous leaf and root traits and sequenced RNA of both leaf and root tissue from a single genotype to: (1) investigate the phenotypic response of sunflower seedlings to each of the fours stresses; (2) characterize the transcriptional response of these seedlings to the various stresses across tissue types; and (3) determine the extent to which these transcriptional responses are shared across stress scenarios.

## Results

### Phenotypic response to stress

In general terms, all stress treatments resulted in relatively poor seedling performance when compared to growth under control conditions (Fig 1A). Each stress scenario resulted in an overall decrease in biomass relative to control; this effect was largely driven by a significant decrease in biomass in response to the dry-down; the other three stresses resulted in biomass values that were intermediate to, but not significantly different from, the control and dry-down scenarios (Table 1). Partitioning total biomass into organ mass fractions, both leaf mass fraction (LMF) and root mass fraction (RMF) differed significantly across treatments (both $P < 0.001$), while there were no significant differences in shoot mass fraction (SMF). Relative to control, LMF was significantly reduced for all stress scenarios, while RMF was increased for all stresses except PEG, which was not significantly different from control. Low-nutrient stress resulted in the largest apparent shift between LMF and RMF (Table 1).

All leaf traits varied significantly across treatments (all $P < 0.05$; Table 1). PEG and salt stress resulted in a significant increase in chlorophyll concentration, whereas PEG, salt, and nutrient stress all increased leaf mass-per-area (LMA). Relative water content (RWC) values were significantly reduced in response to dry-down and salt; the RWC values for PEG and nutrient stress were intermediate to, and not significantly different from, the control and other stress treatments. Leaf element analyses revealed a significant reduction in carbon content under PEG stress, along with a significant increase in $\delta^{13}C$ values in response to all four

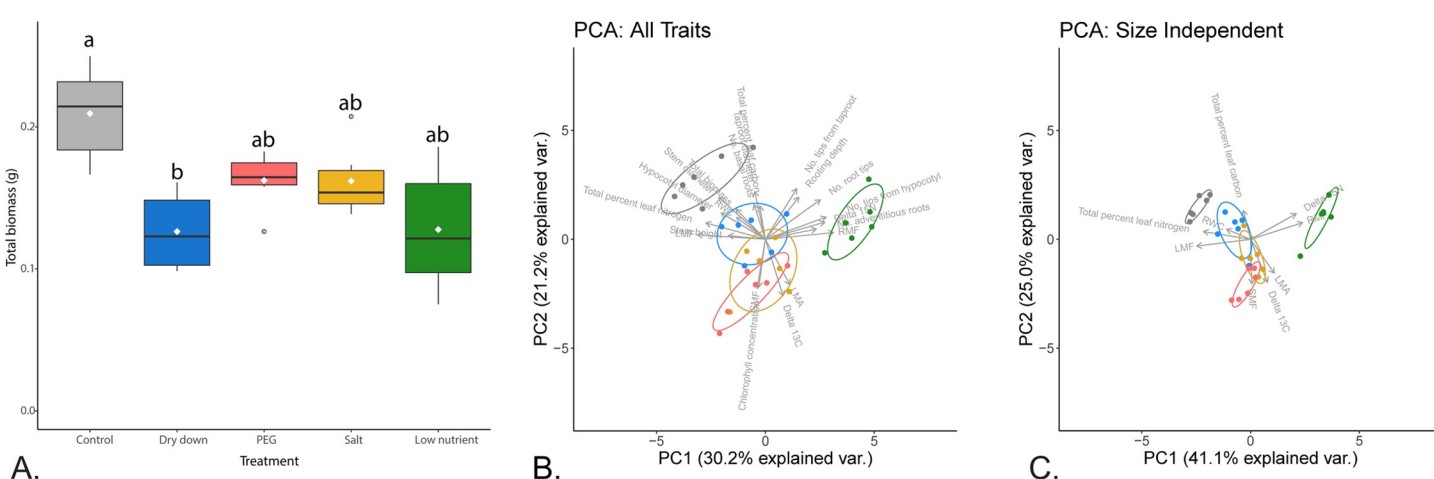

**Fig 1. Phenotypic trait comparison for control vs. stress scenarios.** In all panels, control is shown in gray, dry-down in blue, PEG in red, salt in yellow, and low-nutrient in green. (A) Boxplot of overall plant performance measured as total biomass. Black horizontal bars indicate median, while white diamonds indicate mean values per treatment. Letters above each box correspond to their *post hoc* Wilcoxon groupings. (B) Principal component analysis (PCA) for all measured traits (n = 21) illustrated using the first two PCs. (C) PCA of all size-independent traits (n = 10) illustrated using the first two PCs.

**Table 1. Phenotypic means and standard deviations of all measured traits (n = 21).**

| Traits | Treatment Mean ± Std. Error | | | | |
|---|---|---|---|---|---|
| | Control | Dry-down | PEG | Salt | Low-nutrient |
| Total biomass** | 0.21 ± 0.01[a] | 0.13 ± 0.01[b] | 0.16 ± 0.01[ab] | 0.16 ± 0.01[ab] | 0.13 ± 0.02[ab] |
| Leaf mass fraction (LMF) *** | 0.57 ± 0.01[a] | 0.47 ± 0.02[bc] | 0.49 ± 0.01[b] | 0.44 ± 0.01[c] | 0.21 ± 0.01[d] |
| Root mass fraction (RMF) *** | 0.27 ± 0.02[a] | 0.36 ± 0.02[b] | 0.31 ± 0.02[ab] | 0.37 ± 0.01[b] | 0.6 ± 0.01[c] |
| Stem mass fraction (SMF) | 0.16 ± 0.01[a] | 0.17 ± 0.01[a] | 0.21 ± 0.01[a] | 0.19 ± 0.01[a] | 0.18 ± 0.01[a] |
| Leaf mass per area (LMA) *** | 0.018 ± 0.001[a] | 0.018 ± 0.001[a] | 0.023 ± 0.001[b] | 0.026 ± 0.001[c] | 0.021 ± 0.002[b] |
| Leaf [chlorophyll]*** | 13.35 ± 0.87[a] | 14.02 ± 0.38[a] | 21.54 ± 0.61[b] | 25.13 ± 0.51[c] | 13.12 ± 0.53[a] |
| Relative water content (RWC)** | 0.71 ± 0.1[a] | 0.37 ± 0.04[bc] | 0.49 ± 0.03[ab] | 0.3 ± 0.02[c] | 0.43 ± 0.05[abc] |
| Leaf δ13C*** | -34.74 ± 0.18[a] | -33.15 ± 0.32[b] | -31.78 ± 0.14[c] | -33.03 ± 0.32[b] | -33.46 ± 0.13[b] |
| Leaf δ15N*** | 2.36 ± 0.26[ab] | 2.78 ± 0.13[ac] | 1.49 ± 0.4[b] | 3.24 ± 0.21[c] | 9.51 ± 0.55[d] |
| Total percent leaf carbon* | 38.04 ± 0.44[a] | 37.34 ± 1.31[ab] | 35.34 ± 0.44[b] | 37.43 ± 0.17[a] | 37.18 ± 0.93[ab] |
| Total percent leaf nitrogen*** | 7.77 ± 0.08[a] | 7.39 ± 0.18[ab] | 5.67 ± 0.1[d] | 7.17 ± 0.12[b] | 4.12 ± 0.32[c] |
| Stem height | 90.88 ± 4.96[a] | 71.51 ± 3.59[a] | 85.44 ± 4.85[a] | 78.13 ± 4.3[a] | 73.71 ± 3.38[a] |
| Stem diameter** | 3.38 ± 0.15[a] | 2.73 ± 0.14[ab] | 2.47 ± 0.08[b] | 2.69 ± 0.08[b] | 2.33 ± 0.11[b] |
| Rooting depth | 115.82 ± 7.08[a] | 104.65 ± 7.59[a] | 99.29 ± 10.57[a] | 108.97 ± 7.57[a] | 126.67 ± 3.13[a] |
| No. root tips | 429.33 ± 75.82[a] | 404.17 ± 52.05[a] | 349.67 ± 58.08[a] | 412 ± 42.83[a] | 635.67 ± 61.19[a] |
| No. tips from hypocotyl | 154.33 ± 36.88[a] | 171.33 ± 42.84[a] | 163.5 ± 45.14[a] | 200.67 ± 34.79[a] | 338.67 ± 39.47[a] |
| No. tips from taproot | 182.83 ± 30.49[a] | 161.5 ± 17.04[a] | 125.67 ± 24.65[a] | 138.83 ± 18.55[a] | 208.67 ± 29.04[a] |
| No. adventitious roots^ | 23.5 ± 3.43[a] | 21 ± 5.16[a] | 23.17 ± 5.49[a] | 27.67 ± 6.85[a] | 40.17 ± 3.72[a] |
| No. basal roots | 49.83 ± 4.11[a] | 57.33 ± 6.25[a] | 40.17 ± 2.96[a] | 42 ± 7.39[a] | 40.67 ± 2.5[a] |
| Hypocotyl diameter^ | 0.42 ± 0.02[a] | 0.36 ± 0.02[ab] | 0.35 ± 0.01[ab] | 0.36 ± 0.02[ab] | 0.34 ± 0.01[b] |
| Taproot diameter | 0.29 ± 0.01[a] | 0.26 ± 0.01[a] | 0.25 ± 0.01[a] | 0.26 ± 0.01[a] | 0.27 ± 0.01[a] |

Superscript of asterisk and/or caret indicate significance of ANOVA effects, while letters indicate the *post-hoc* Wilcoxon groups. Significance for treatment effects indicated as:

*** $P < 0.0001$,

** $P < 0.001$,

* $P < 0.05$. Significant block effects are denoted with a "^".

stresses. The PEG-stressed individuals had the most extreme change relative to the control. Leaf nitrogen content was significantly reduced under PEG, salt, and nutrient stress, whereas $\delta^{15}N$ was significantly increased under salt and (especially) nutrient stress. Stem diameter varied significantly across treatments ($P < 0.01$) but stem height did not (Table 1). PEG, salt, and nutrient stress all reduced stem diameter while dry-down produced an intermediate value that was not significantly different from control or the other stresses. Interestingly, despite being the tissue that interacted most directly with each of the stresses, none of the root traits exhibit significant treatment effects (all $P > 0.06$) except RMF (as described above) and hypocotyl diameter, which also had a significant block effect.

To visualize the overall phenotypic response of seedlings to the various treatments, a principal component analysis (PCA) of all 21 traits was conducted. The first two principal components accounted for 51.4% of the phenotypic variance explained (Fig 1B). Here, LMF, RMF, no. tips from the hypocotyl, total percent leaf nitrogen, and $\delta^{15}N$ loaded the most strongly on PC1 (S1 Table), which explained 30.2% of the observed variation and largely accounted for the separation of nutrient stress from the remaining treatments. $\delta^{13}C$, no. tips from the taproot, shoot mass fraction (SMF), chlorophyll concentration, and taproot diameter loaded most strongly on PC2. This axis accounted for 21.2% of the observed variation, and primarily reflected the separation of the stress treatments from control. The phenotypic responses to the

various treatments were also visualized using the size-independent traits to determine if observed similarities held up in the absence of metrics reflecting growth or overall performance (Fig 1C). In this case, RWC, total percent leaf nitrogen, LMF, δ15N, and RMF loaded most strongly on PC1 (41.1% of variance explained), which primarily separated nutrient stress and control from the remaining treatments (Fig 1C). In contrast, total percent leaf carbon, SMF, δ13C, and LMA loaded most strongly on PC2 (25.0% of variance explained), which separated dry-down and PEG stress, with salt stress falling intermediate to and overlapping with those two stresses.

## Transcriptomic analysis and patterns of differential expression in response to stress

We sequenced RNA from 38 samples across the five treatments (control + four stresses) and two tissue types (leaf and root). We removed 5 outlier libraries with abnormal gene expression patterns based upon multidimensional scaling (MDS) plots generated from the expression data (S1A and S1B Fig); the remaining 33 libraries averaged 15 million paired-end reads per library. Reads generated from RNA sequencing mapped to 39,042 unique genes across all five treatments and both tissue types. We then calculated sets of DEGs between tissues, between control samples and all stresses combined, and between control samples and each stress for leaf and root tissue individually (S1 Table). When identifying DEGs between all stresses in combination with the control, there were significantly more DEGs found in leaf tissue than in root tissue (leaf = 9,317; root = 7,412; $P < 0.001$; $\chi^2$ test; Table 2), and in both tissues there were more downregulated genes than upregulated genes. When identifying DEGs for each stress individually, a total of 22,915 unique genes were differentially expressed; 8,754 DEGs were unique to leaf tissue and 6,279 DEGs were unique to root tissue. Nutrient stress resulted in the largest number of DEGs in leaf tissue while PEG stress resulted in the most DEGs in root tissue. Of the three water-related stresses, PEG had the largest number of DEGs followed by salt stress and dry-down; a pattern consistent across tissue types. Each stress had significantly more DEGs in leaf tissue than in root tissue (Table 2; all $P < 0.001$; $\chi^2$ test). Nutrient stress had the greatest number and largest proportion of unique DEGs in both leaf and root tissue while dry-down stress had the fewest unique DEGs and the lowest proportion of unique DEGs (Table 2 and Fig 2). In leaf tissue, all stresses except dry-down resulted in more upregulated DEGs than downregulated DEGs. This pattern does not remain consistent in root tissue, as dry-down and PEG stress resulted in more upregulated vs. downregulated DEGs, while salt and nutrient stress had more downregulated vs. upregulated DEGs.

**Table 2. The number of DEGs and direction of change as compared to control in each tissue for each stress individually and all stresses combined.**

| | Stress | # of DEGs | # upregulated | # downregulated | DEGs unique to stress |
|---|---|---|---|---|---|
| **Leaf** | Dry-down | 1384 | 578 | 806 | 152 (10.98%) |
| | PEG | 9686 | 4846 | 4840 | 2040 (21.06%) |
| | Salt | 8351 | 4351 | 4000 | 1827 (21.88%) |
| | Low-nutrient | 11059 | 5608 | 5451 | 3644 (32.95%) |
| | All stresses combined | 9317 | 4049 | 5268 | NA |
| **Root** | Dry-down | 717 | 504 | 213 | 96 (13.39%) |
| | PEG | 8624 | 4607 | 4107 | 3201 (37.12%) |
| | Salt | 5727 | 2838 | 2889 | 1487 (25.96%) |
| | Low-nutrient | 7527 | 3514 | 4013 | 3280 (43.58%) |
| | All stresses combined | 7412 | 3076 | 4336 | NA |

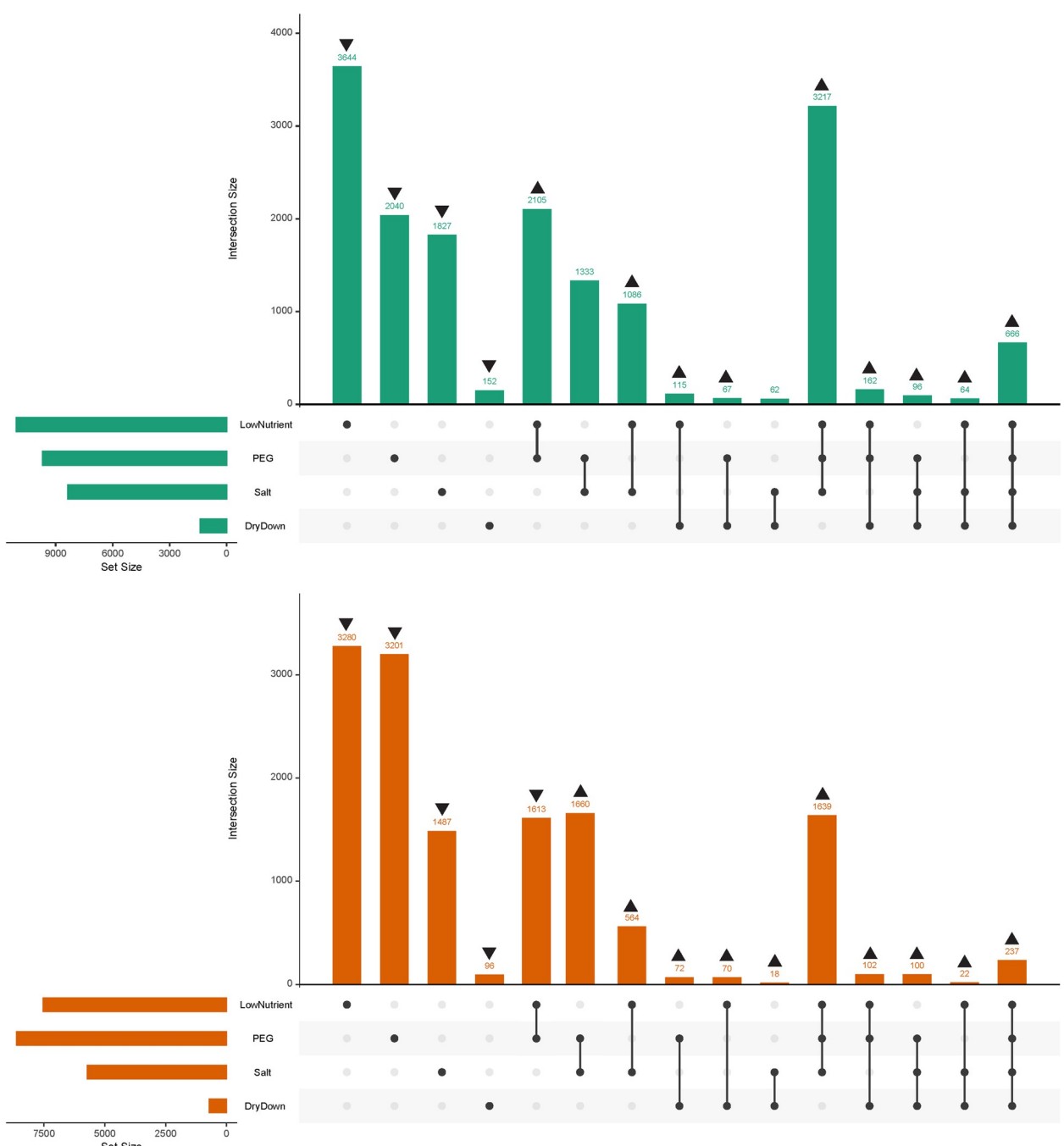

**Fig 2. UpSet plot depicting the number of unique DEGs shared among combinations of stresses.** DEGs shared between stresses are depicted by filled black dots in each stress category with connecting lines between them. For categories with only a single stress, the number of DEGs depicted are only those unique to that individual stress and do not contain DEGs that are shared between multiple stresses. Up and down arrows above sets represent significantly more or less DEGs than expected by chance, respectively. The total number of DEGs for a given stress is represented by the histogram in the lower left portion of each panel. (A) Shared DEGs in leaf tissue (green). (B) Shared DEGs in root tissue (orange).

The number of DEGs unique to each individual stress was significantly lower than expected by chance in both leaf and root tissue (Fig 2 and S2 Table). Consequently, most combinations of stress had significantly more shared DEGs than expected by chance, with the exception of

the PEG + salt stress and salt + dry-down stress combinations in leaf tissue, which contained the expected number of shared DEGs, and the nutrient + PEG stress combination in roots, which contained significantly fewer shared DEGs than expected by chance. In both leaf and root tissue there were more DEGs shared among all four stresses than there were shared among just the three water-related stresses. Stress combinations that included dry-down tended to have fewer shared DEGs than stress combinations that excluded the dry-down stress; however, this was simply a consequence of the dry-down stress having the smallest overall number of DEGs in both tissues. Most genes differentially expressed in multiple stresses were expressed in the same direction in response to each stress. In both leaf and root tissue, the dry-down/salt/nutrient stress (DSN) and dry-down/salt stress (DS) combinations had noticeably higher proportions of differentially regulated DEGs (i.e., DEGs that exhibited a mix of up- and down-regulation across stresses) than the other stress combinations. This was also true for the salt/nutrient (SN) stress combination in root tissue (Table 3).

To visualize the overall differences in the transcriptomic response to the various treatments, multidimensional scaling (MDS) analysis was conducted on the whole leaf and root transcriptome (Fig 3A and 3B) and using just the DEGs that were shared across all stresses in each tissue (Fig 3C and 3D). MDS analyses of the whole transcriptome in leaf and root tissue revealed that nutrient stress had a very distinctive transcriptome profile as compared to the three water-related stresses and controls (Fig 3A and 3B). PEG and salt stress elicited similar responses,

**Table 3. Directionality of DEG expression among stress intersections.**

|      | Stress combination | # upregulated | # downregulated | # differentially regulated | % differentially regulated |
|------|--------------------|---------------|-----------------|----------------------------|----------------------------|
| **Leaf** | DSPN | 390 | 272 | 4 | 0.60% |
|      | DSP | 29 | 59 | 8 | 8.33% |
|      | DSN | 26 | 27 | 11 | 17.19% |
|      | DPN | 126 | 34 | 2 | 1.23% |
|      | SPN | 1646 | 1521 | 50 | 1.55% |
|      | DS | 24 | 21 | 17 | 27.42% |
|      | DP | 48 | 19 | 0 | 0.00% |
|      | DN | 41 | 62 | 12 | 10.43% |
|      | SP | 573 | 758 | 2 | 0.15% |
|      | SN | 449 | 577 | 60 | 5.52% |
|      | PN | 991 | 1068 | 46 | 2.19% |
| **Root** | DSPN | 68 | 162 | 7 | 2.95% |
|      | DSP | 15 | 82 | 3 | 3.00% |
|      | DSN | 14 | 4 | 4 | 18.18% |
|      | DPN | 33 | 64 | 5 | 4.90% |
|      | SPN | 844 | 698 | 97 | 5.92% |
|      | DS | 4 | 5 | 9 | 50.00% |
|      | DP | 8 | 64 | 0 | 0.00% |
|      | DN | 37 | 28 | 5 | 7.14% |
|      | SP | 738 | 918 | 4 | 0.24% |
|      | SN | 253 | 167 | 144 | 25.53% |
|      | PN | 906 | 651 | 56 | 3.47% |

Upregulated DEGs show increased expression under each stress scenario. Downregulated DEGs show lower expression under each stress scenario. Differentially regulated DEGs are upregulated in at least one stress and downregulated in at least one stress within a combination of stresses. The % of differentially regulated DEGs column displays the percentage of DEGs differentially regulated among all DEGs for a given stress combination. D = dry-down stress, S = salt stress, P = PEG stress, N = low-nutrient stress.

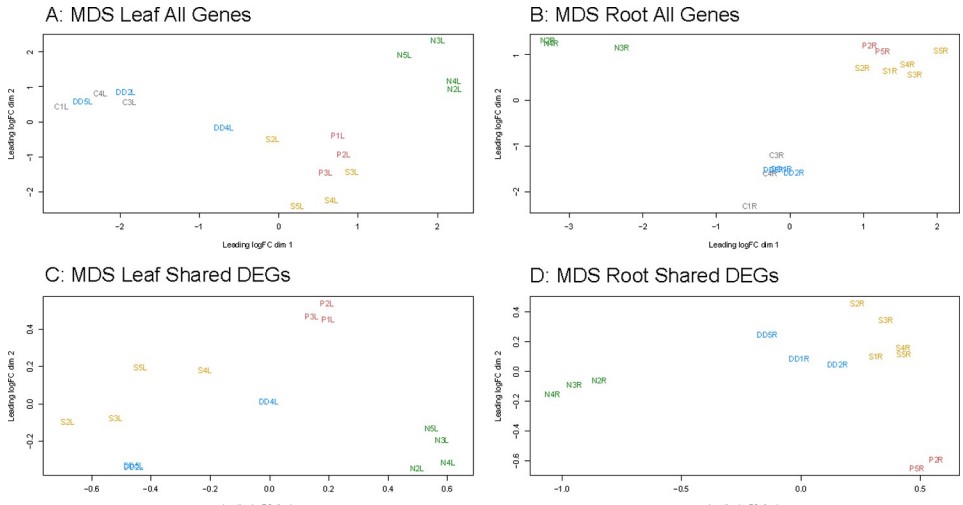

**Fig 3. MDS plots illustrating the transcriptomic response to each stress across samples.** (A) MDS plot based on the expression values of all genes included in this analysis for leaf tissue and (B) root tissue. (C) MDS plots generated from the expression values of DEGs shared among all four stresses for leaf tissue and (D) root tissue. Control samples were not included in panels C and D as the genes under consideration in these analyses were by definition differentially expressed relative to control.

with those samples tending to cluster together in both leaf and root tissue. The dry-down stress samples behave similarly to the control samples, consistent with the observation that dry-down resulted in the fewest DEGs of all the stresses in both tissues. A similar pattern is observed when conducting MDS on only the DEGs shared between all stresses (Fig 3C and 3D). We elected to remove the control samples from these latter plots as the DEGs by definition are significantly different between the stress and control samples. Nutrient stress forms its own distinctive group separated from the three water-related stresses in both tissues. Unlike the whole transcriptome MDS, salt and dry-down cluster close together while PEG stress elicits a more divergent transcriptomic response.

To further investigate the putative function of DEGs found under each stress treatment, we conducted GO term enrichment analyses for all sets of DEGs, including the full set of DEGs for all stresses combined in leaf and root tissue, the shared DEGs between leaf and root tissue for all stresses combined, the unique DEGs in leaf vs. root tissue for all stresses combined, each stress individually in leaf and root tissue, and all sets of unique shared DEGs for each stress combination in leaf and root tissue (S3 Table). Several GO terms were found to be significantly enriched within the set of DEGs for most stress and tissue combinations. However, there were no significantly enriched GO terms among DEGs shared by all four stresses in leaf tissue and only a single significantly enriched GO term among DEGs shared by all four stresses in root tissue (GO:0006633 fatty acid biosynthetic process). The significant GO terms enriched among the set of DEGs shared between the three water related stresses in root tissue were GO:0004724 magnesium-dependent protein serine/threonine phosphatase activity and GO:0006470 protein dephosphorylation; there were no significantly enriched GO terms shared by the three water-related stresses in leaf tissue.

We also conducted a KEGG enrichment analysis to determine if any sets of shared DEGs belonged to any specific metabolic pathways (S4 Table). The set of DEGs from all stresses combined in root tissue had 96 significantly enriched KEGG terms, which was the most of any DEG set. Of those 96 enriched KEGG terms, 36 of them belonged to KEGG map01100 'Metabolic pathways' while 22 of them belonged to map01110 'Biosynthesis of secondary

metabolites' pathway. No other pathway had double-digit enriched KEGG terms from our set of root DEGs. Of the enriched KEGG terms, 59 were terms from the 'Enzyme' class and 16 were from the 'Transporter' class. Few other DEG sets had a notable number of enriched KEGG terms. However, the KEGG term KO9872 'aquaporin PIP' was enriched in the three sets of DEGs for PEG, salt, and nutrient stress in root tissue, but not in the set of DEGs for dry-down stress in root tissue.

### Gene co-expression network analysis

We constructed a signed gene co-expression network to identify modules of genes specifically associated with each tissue and stress combination (Fig 4). We identified 30 co-expression modules ranging from 31 to 18,522 genes in each module (the minimum module size for this analysis was set to the default value of 30). We found that 9 modules were significantly correlated with all stresses in leaf tissue and 10 modules were correlated with all stresses in root tissue (Fig 4).

The only module that was significantly correlated with more than one stress/tissue combination was the 'greenyellow' module, which was significantly positively correlated with both the PEG and salt treatment in leaves. A positive correlation between a module and a stress in this case means that the genes within the module are all upregulated under the stress condition. The 'cyan' and 'pink' modules were significantly correlated with nutrient stress in leaf and root tissue, respectively. However, the 'cyan' and 'pink' modules were not significantly associated with tissue-specific expression and were weakly correlated with nutrient stress in the tissue type for which they were not significant. The 'yellow' module was also significantly associated with the dry-down stress in root tissue and this is the second largest module associated with any given stress containing 2,318 genes. Most other modules significantly correlated with a given stress/tissue combination contained fewer than 600 genes, with the exception of the 'green' and 'turquoise' modules, which contained 2,231 and 18,522 genes and were significantly correlated with nutrient stress in leaf tissue and root tissue, respectively. The large size of these two nutrient-related co-expression modules is consistent with the observation that nutrient stress elicited the broadest transcriptional analysis of all stresses considered, consistent with the results of our differential expression analysis.

We also tested for module association with control samples and found a significant positive correlation between control leaf tissue samples and the 'salmon' and 'royalblue' modules; as expected, these two modules were not correlated with any stress/tissue combination (S2 Fig). Lastly, the 'grey' module is significantly associated with dry-down stress in root tissue; however, WGCNA puts all genes that are not significantly co-expressed with any other genes into 'grey'. Therefore 'grey' is not considered to be a true co-expression module and no biological interpretation should be made relating to this 'module' (Fig 4).

To explore the putative functions of genes contained with co-expression modules, we tested for GO and KEGG term enrichment in each module; however, many modules were not significantly enriched for any GO terms after multiple hypothesis correction, especially those modules that had relatively few genes (S5 and S6 Tables). The 'greenyellow' module, which was significantly correlated with both PEG and salt stress in leaf tissue, was significantly enriched for GO:0006869 lipid transport and GO:0008289 lipid binding. The 'yellow' module, significantly correlated with dry-down stress in root tissue, was enriched for 49 GO terms which included GO:0006833 water transport, several cell wall reorganization related terms, and more GO terms related to fatty acid metabolism. The 'green' and 'turquoise' modules, which were associated with nutrient stress in leaf and root tissue, respectively, did not share any enriched

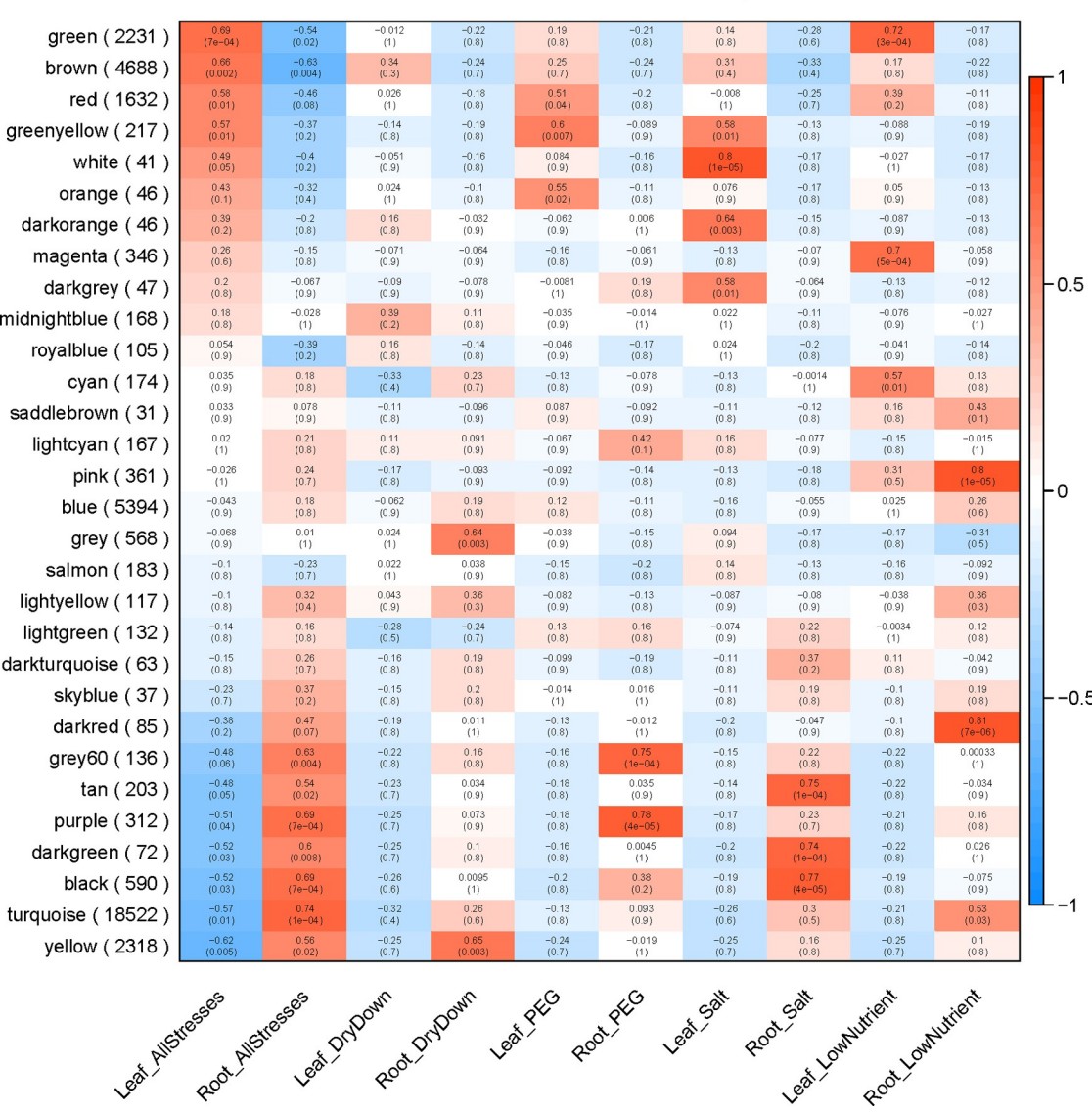

**Fig 4. Visual depiction of correlations between gene co-expression modules and tissue/treatment.** Correlation values (upper text) and *P*-values (lower, parenthetical text) are presented in each cell. Color is determined by the sign and magnitude of the correlation. Positive correlations (red) indicate genes within a module are upregulated within a stress/tissue combination while negative correlations (blue) indicate that genes within the module are downregulated.

GO terms despite their large size, indicating a truly disparate transcriptomic response between the two tissues. The 'blue' module was significantly enriched for seven KEGG terms belonging to the 'Chromosome and associated proteins' KEGG category. Those terms included several different kinds of histones, suggesting that this module may have something to do with chromosome organization. However, the blue module was not significantly associated with any stress or tissue combination. No other module was significantly enriched for sets of KEGG terms that would imply that module was associated with a specific metabolic pathway. In general terms, modules significantly correlated with each stress/tissue combination had non-overlapping sets of enriched GO terms, highlighting the uniqueness of the transcriptomic response for each of the four stresses and across tissue types.

## Comparison of differential expression analysis to the co-expression network

To compare the results of the differential expression analysis with the co-expression network, we asked if more DEGs from the sets of DEGs shared between stresses belonged to specific modules than expected by chance by testing for deviations from expected module membership using 1,000 simulated networks (S7 Table). In leaf tissue, no set of DEGs had significantly more DEGs in a module than expected by chance, though there were 17 module/intersection combinations with *P*-values equal to ~0.053 after multiple hypothesis correction. In root tissue, there were 25 module/intersection combinations that had significantly more DEGs in a module than expected by chance after multiple hypothesis correction (S7 Table).

The 'brown' and 'purple' modules were both significantly correlated with all stresses in leaf tissue and there were significantly more genes than expected by chance that were differentially expressed in leaf tissue for the intersection of all four stresses. In the root tissue intersection of all four stresses, significantly more DEGs were found belonging to the 'darkgreen' module than were expected by chance. Among DEGs that were unique to a single stress in root tissue and modules that were significantly correlated with that specific stress in the network, more DEGs than expected by chance were found in the 'darkgreen' and 'black' modules for salt stress and in the 'turquoise' module for nutrient stress. For DEGs unique to the intersection of the three water-related stress, more DEGs were found in the 'green', 'grey60', and 'purple' modules than expected by chance, however, only the 'purple' and 'grey60' modules were significantly correlated with a specific stress in the network and both modules were only correlated with PEG stress.

## Discussion

We observed substantial variation in the phenotypic response to the four stresses investigated herein. For all stress treatments, there was a generalized reduction in biomass, although this was only significant under dry-down stress. Furthermore, we also observed an increase in RMF along with a decrease in LMF for all four stresses, though for PEG the increased allocation to roots was not significantly different from the control (Table 1). Our data are thus consistent with previous studies highlighting a trade-off between leaf and root mass fractions under stress [e.g., 95–99]. For the low-nutrient treatment, this shift between above and belowground biomass was more pronounced, with a majority of total biomass accumulation under low-nutrient conditions occurring belowground (Table 1). While such an increase in RMF can be caused by allometric scaling (i.e., smaller plants have a larger proportion of root biomass; [29,30,32,100,101], this scaling relationship may be functional, with smaller (stressed) individuals preferentially allocating more biomass to roots [102]. Consistent with this idea, our data revealed an absolute increase in the number of root tips and adventitious roots under nutrient stress, though these differences were not significant (Table 1).

Overall, our results highlight a clear separation in phenotypic response between the three water-related stresses and the nutrient stress relative to the control (Fig 1B and 1C). The separation of the nutrient stress samples from the water-related stresses and control samples in trait space is predominantly driven by traits such as RMF, leaf δ15N, number of root tips from the hypocotyl, and number of adventitious roots (Fig 1B). Interestingly, this separation remains apparent in the absence of size-dependent traits indicating that it is not simply a byproduct of differences in vigor. Rather, traits such as RMF and δ15N continue to drive the difference between the nutrient samples and all others (Fig 1C). In this context, it should be noted that the nutrient stress was the longest lasting of all the stress treatments during our experiment, as these seedlings were never provided with supplemental nutrients. While the

PEG and salt stresses were likewise continuous, they were not implemented until after seedling establishment. Nonetheless, the nutrient stress exhibited a remarkably different phenotypic (and ultimately transcriptomic) difference from the other three stresses.

It is also worth noting that phenotypic response to the three water-related stresses could be driven by multiple factors, including potential differences in stress severity, differences in the mode of implementation (i.e., cyclical vs. continuous), and/or the nature of the stress itself (i.e., true water limitation vs. an osmotic challenge vs. ion toxicity). Indeed, though PEG is touted as a drought mimic [4], it results in a phenotypic shift that more closely resembles salt vs. dry-down stress (Fig 1B and 1C). While the short-term osmotic effects of both PEG and salt stress can resemble those of water deficits due to soil drying, longer-term responses are likely to be more related to the osmotic effects of PEG vs. the toxic effects of NaCl [4,38,91,103,104]. Moreover, both of these treatments were maintained at a fixed level throughout the experiments, as opposed to the episodic nature of periodic dry-downs. Despite these differences, we found that the observed phenotypic differentiation between all water-related stress treatments and the control was largely driven by variation in aboveground traits, particularly those related to leaf physiology (Fig 1B and 1C). Here, seedlings under stress experienced trait shifts commonly found under low-resource conditions [23,102,105], including higher (i.e., less negative) carbon fractionation values and leaf C:N ratios for all water-related stresses [106,107], two traits that may indicate a reduction in photosynthetic rate [23]. The observed increase in chlorophyll was likely due to an increase in LMA, which is known to produce elevated values due to increased leaf thickness [108]; Table 1).

In terms of the transcriptomic response to stress, we observed substantial changes in gene expression in both leaf and root tissues with nearly half of all expressed genes being differentially expressed in response to at least one stress in roots and/or leaves. Nutrient stress resulted in the largest number of DEGs in leaf tissue and the second largest number in roots, consistent with its major effect on both LMF and RMF. In roots, PEG stress resulted in the greatest number of DEGs despite little phenotypic differentiation of root traits between PEG stressed and control samples. Comparison of the PCAs based on phenotypic and transcriptomic data showed similar overall patterns; specifically, the response to nutrient stress was different (and more pronounced) when compared to the three other stresses in terms of both phenotypic and transcriptomic variation.

The relatively small differences in root-related responses between treatments (Table 1) is consistent with the fact that there tended to be fewer DEGs in roots unique to each individual stress than expected by chance. Indeed, DEGs in roots tended to be shared by multiple stresses indicating at least a somewhat shared transcriptomic response to stress. As the roots were in direct contact with each stress treatment (the presence of salt or PEG or the absence of sufficient nutrients or water), it is within reason to expect that roots would have a greater proportion of shared DEGs than leaf tissue. In addition, the majority of shared DEGs had the same directionality of expression across stresses (Table 3). However, in both leaf and root tissue the dry-down/salt/nutrient and dry-down/salt stress combinations had larger proportions of DEGs with different directionalities of expression. Furthermore, our KEGG analysis of DEGs indicated that this partially shared transcriptomic response in roots may involve the biosynthesis of secondary metabolites and the production of PIP aquaporins, though it is important to note that there was not an enrichment for DEGs related to PIP aquaporins in the dry-down treatment.

Network modules were only ever significantly correlated with a single stress in root tissue, suggesting each stress resulted in a fairly unique transcriptomic response in roots. This is not necessarily inconsistent with the finding that large numbers of DEGs were shared between multiple stresses; rather, it is an indication that the transcriptomic response to stresses is

complex and likely includes both genes that are part of a universal stress response pathway and genes that respond only under specific stress conditions. Furthermore, the lack of significant differences between root traits under each stress treatment is not indicative of the lack of a unique response to each stress in root tissue, as clearly shown through our transcriptomics data. Rather, similar phenotypic responses may be achieved through different means. There is also a possibility that the more unique aspects of the root transcriptomic response to each stress may only manifest phenotypically after continued imposition of stress beyond the seedling stage or in a natural environment.

The relationship between the phenotypic and transcriptomic responses to stress in leaves is more apparent. Even though there are fewer DEGs in leaf tissue unique to each individual stress than expected by chance, there are still large numbers of unique DEGs which could result in unique phenotypic changes. Furthermore, some leaf phenotypic changes were shared between multiple stresses. For example, an increase in chlorophyll content was detected under both PEG and salt stress. While there were no more or fewer shared DEGs than expected by chance between PEG and salt stress in leaf tissue, the only network module to be significantly correlated with more than one stress/tissue combination (i.e., greenyellow) was significantly correlated with PEG and salt stress in leaf tissue, suggesting a strong shared transcriptomic response between these two stresses. Other phenotypic changes shared between multiple stresses included increased LMA among PEG, salt, and nutrient stress (there was likewise a large set of shared DEGs between those stresses in leaf tissue) and RWC for salt and dry-down stress, which did not deviate from the expected number of shared DEGs.

Although dry-down stress resulted in the largest overall decrease in biomass of all four stresses, it had the smallest number of DEGs in both tissues. The results of the DEG analysis are supported by the MDS plots as dry-down stress and control occupy the same space when the expression of all genes are considered (Fig 3A and 3B). Dry-down stress is seemingly the most severe phenotypically yet elicits the smallest changes to the transcriptome.

Importantly, our results also indicate that PEG stress is not a suitable substitute for drought stress–at least not as implemented here as a periodic dry-down. Rather, the transcriptomic response to PEG stress was very similar to that of salt stress while the phenotypic response was also quite different from dry-down (Table 1). PEG stress is known to both reduce the ability for roots to uptake oxygen [109] and also interfere with the uptake of sodium and potassium [110]. The ability of PEG to limit sodium salt uptake may explain why PEG and salt stress have a similar transcriptomic response. PEG does enter into plant tissues and travel through the apoplast to affect membrane transport sites similarly to salt [110]. As such, caution should be exercised when attempting to mimic drought conditions using an osmoticum.

## Conclusions

A comparison of the phenotypic data, differential expression analysis, and gene co-expression network analysis revealed that these approaches to assessing an abiotic stress response can produce results that are seemingly at odds with each other. Significant changes to phenotypes under stress are not necessarily reflected by large changes in the transcriptome as evidenced by the dry-down stress treatment. Furthermore, while the differential expression analysis indicated that large numbers of DEGs were shared between multiple stresses, the co-expression network analysis provided a more nuanced view and suggested that the response to these stresses is mostly due to non-overlapping sets of co-expressed genes. Indeed, despite the sizable collection of shared DEGs, only a single co-expression module was significantly correlated with more than one stress, and no modules were significantly correlated with the same stress in both tissues. While there is some overlap in the transcriptomic response to these four

stresses at the level of individual genes, the situation is considerably more complex once the correlation structure is taken into account. Ultimately, the observed differences between the results of the phenotyping, differential expression analysis, and gene co-expression analysis highlights the utility of taking a more pluralistic approach to the analysis and interpretation of RNAseq data.

## Methods

### Plant materials, seedling establishment, and experimental design

Seeds from a single, inbred oilseed sunflower line (HA412-HO; PI 642777) were obtained from the North Central Regional Plant Introduction Station (NCRPIS) in Ames, IA. Seeds were planted 1.5 cm deep in individual 50 mL Falcon tubes that had been pre-drilled with two ⅛-inch holes 2 cm from the base to allow drainage. The growth substrate was a mixture of sand and Turface (3:1, v/v; Oldcastle APG Northeast, Inc., Manassas, VA). The tubes were then placed in a plant growth room where the seedlings were maintained for 20 days following germination under one of 5 differential treatments (i.e., control plus 4 stress treatments; see below). Throughout the experiment, the temperature was kept at 20°C with a 16h:8h day:night cycle. All individuals were arranged in a randomized block design with two blocks and 11 biological replicates per treatment (5–6 per block; n = 55 seedlings total). Six replicates per treatment (three per block) were randomly designated for destructive phenotypic analyses. The remaining five replicates per treatment (2–3 per block) were designated for transcriptomic profiling, with all leaf and root tissues being harvested separately, frozen in liquid nitrogen, and stored at -80°C prior to RNA extraction.

### Treatment implementation

Upon planting, the control plants plus all individuals from the three water-related stresses were top-watered daily with a solution of deionized (DI) water and supplemental nutrients in the form of one g/L of Jack's All Purpose 20-20-20 aqueous mix (J.R. Peters, Inc., Allentown, PA) for 10 days to facilitate seedling establishment. Following establishment, the control seedlings were maintained as above. The water-related stresses were implemented at the V2 stage of sunflower development [111] as a repeated dry-down to mimic drought stress through top-down soil drying, an osmotic challenge to limit water uptake using polyethylene glycol (PEG-6000, 8.25% by volume, which is sufficient to induce an osmotic challenge of -0.25 MPa; [112]), and salt (NaCl, 100 mM) stress. In the repeated dry-down, the seedlings were transitioned from daily watering to watering with the control solution on alternate days. This decrease in frequency was sufficient to induce visible symptoms of water limitation (i.e., the seedlings began to wilt prior to re-watering) without causing mortality.

In the PEG and salt scenarios, the seedlings were transitioned to watering with treatment solutions containing the appropriate amount of PEG or salt with one g/L of Jack's All Purpose 20-20-20 aqueous solution dissolved in DI water. All individuals were top-watered with their respective treatment solutions to bring their growth substrate to full capacity in 50mL cone-shaped containers. Osmotic stress, particularly when mediated via high molecular weight polymers such as PEG, has been a generally accepted approach for inducing water limitation in a uniform and repeatable way [e.g., 92,93,113–118] while minimizing toxicity effects [4,40], but see [91]). In contrast, NaCl not only influences water uptake, but also has the potential to enter cellular pores and elicit toxic effects [91,103,104]. These differential treatments were maintained for 10 days following establishment, for a total of 20 days of seedling growth.

The individuals subjected to low-nutrient stress were top-watered daily with DI water to bring their growth substrate to full capacity, but were not provided with supplemental

nutrients at any time during the experiment. This resulted in a generalized nutrient deficit relative to control conditions, similar to what might be encountered in highly degraded soils limited in nitrogen, phosphorus, calcium, or other micronutrients. Previous work has shown that this low-nutrient treatment limits growth in sunflower, but does not prevent them from completing their lifecycle [119].

## Phenotypic measurements

A total of 21 morphological and physiological traits were measured for each phenotyped individual at the V2 stage of sunflower development after each treatment. As detailed below, these included leaf, stem, and root traits, as well as overall biomass production.

**Overall plant performance.** Total biomass, which served as an integrated metric of overall plant performance, was calculated as the sum of the dried leaf, stem, and root tissue (see below). Organ biomass fractions were also determined from these data as proportions of total biomass.

**Leaf traits.** Ten days after the start of the experiment, corresponding to initiation of the water-related stress treatments, the most recently developed leaf pair was tagged with string. Upon harvest, the next higher leaf pair was designated for measurements. This approach ensured that the leaves of interest were produced following the implementation of the stresses. One leaf from this pair was used to estimate chlorophyll concentration and relative water content; the other leaf was removed and scanned with a CanoScan 8800F scanner (Canon USA, Inc., Melville, NY) at 600 dpi, and dried for biomass estimation and isotope analyses.

An *in situ* optical measure of chlorophyll concentration per unit leaf area was assessed using an Apogee MC-100 chlorophyll content meter (Apogee Instruments, Logan, UT). Two readings were taken for each leaf from different parts of the leaf lamina and averaged. A leaf disc was taken using a ¼-inch diameter hole punch and used to assess leaf relative water content (RWC), which serves as an indicator of plant water status [26]. This was calculated as (FM-DM)/(HM-DM), where FM was fresh mass at the time of collection, HM was hydrated mass (estimated after hydrating the leaf punches for 24 hours), and DM was dry mass, estimated after drying the leaf punches at 60˚C for 72 hours in a forced-air drying oven [120].

Leaf area estimates were obtained from the scanned images using ImageJ [121]. After scanning, the leaves were dried as above and weighed to estimate dry biomass, which was used to calculate leaf mass per area (LMA). The dried leaves were then ground into a fine, homogenous powder using a Thomas Model 4 Wiley ball mill (Thomas Scientific, Swedesboro, NJ) for stable isotope analyses, which were performed at the University of Georgia's Stable Isotope Ecology Laboratory (http://siel.uga.edu/). This yielded estimates of total percent leaf carbon and leaf nitrogen, as well as carbon and nitrogen isotope composition ($\delta^{13}C$, $\delta^{15}N$).

All remaining leaves were dried as above and weighed, and all leaf weights for a given seedling were summed to provide an overall estimate of leaf biomass. Leaf mass fraction (LMF) was then calculated as leaf biomass divided by total biomass.

**Stem traits.** After harvest, stem height and diameter were measured using Fowler 6"/150mm Ultra-Cal IV Electronic Calipers (Fowler Tools and Instruments, Newton, MA). Stem diameter was measured just above soil level. Stem tissue was then dried as above and weighed to estimate stem biomass. Stem mass fraction (SMF) was then calculated as stem biomass divided by total biomass.

**Root traits.** Seedlings were gently uprooted, and root tissue was rinsed to remove soil substrate. Intact roots were patted dry, fanned out, and placed in a vertical orientation on a matte black cloth for imaging. A coin (US penny; 19.05 mm diameter) was placed next to each root system for scale. Photos were then taken with a 12 megapixel camera from a fixed distance of

175 mm and uploaded to the Digital Imaging of Root Traits (DIRT) pipeline [122] with a masking threshold calibration of 5.00. The following traits were assessed: number of tip paths and rooting depth skeleton (both are "common traits" in DIRT); roots seg 1 and 2, number of adventitious and basal roots, hypocotyl diameter, and taproot diameter (all of these are "dicot traits" in DIRT). Following imaging, the roots were dried as above and weighed to estimate root biomass. Root mass fraction (RMF) was then calculated as root biomass divided by total biomass.

## Statistical analysis of phenotypic traits

All analyses of phenotypic data were performed in R v4.04 [123]. To protect against violations of the assumption of normality, a nonparametric Kruskal-Wallis test used to test for an overall treatment effect while controlling for block effects. A pairwise Wilcoxon test with an FDR *P*-adjustment method [124] was used to test for differences in phenotypic responses between stress treatments for all traits with a significant ($P < 0.05$) treatment effect. Finally, two principal component analyses (PCAs) were performed; one of these analyses used the full set of traits (n = 21) to assess the clustering of treatments across trait space, and the other used just the size-independent traits (n = 10) to capture treatment clustering without the influence of metrics related to overall growth and performance. The bioconductor package pcaMethods [125] was used to impute values for 11 missing data points out of 630 individual measurements.

## RNA extraction and sequencing

Tissue samples from five biological replicates for each of the five treatments were used for total RNA isolation, with replicates maintained as separate samples (i.e., they were not pooled). Leaf and root tissue were ground separately in liquid nitrogen using a pre-chilled mortar and pestle to produce a fine powder (ca. 100 μg per sample). Total RNA was then extracted from the ground samples using the RNeasy Mini Kit (Qiagen, Inc., Germantown, MD). RNA extractions were treated to remove DNA contamination using a TURBO DNA-free kit (ThermoFisher Scientific, Waltham, MA). The quality and quantity of each RNA sample was assessed using a NanoDrop 2000 (ThermoFisher Scientific) and an Agilent 2100 Bioanalyzer (Agilent Technologies, Alpharetta, GA). Only RNA samples with 260/280 ratios from 1.8 to 2.1, 260/230 ratios $\geq$ 2.0, and RNA integrity number (RIN) values greater than 7.5 were used for subsequent analyses. Approximately 1 μg of total RNA from each sample was used to construct sequencing libraries using the KAPA Stranded mRNA-Seq Kit (KAPA Biosystems, Wilmington, MA). Thirty-eight individual libraries passing our quality control standards were generated (control: four leaf, three root; dry-down: four leaf, three root; PEG: four leaf, three root; salt: five leaf, five root; low-nutrient: four leaf, three root) and sequenced (paired-end, 75 bp reads) at the Georgia Genomics and Bioinformatics Core (http://dna.uga.edu/) on an Illumina NextSeq (Illumina, San Diego, CA).

## Sequence assembly, read mapping, and gene expression analyses

RNAseq data was processed using a custom bioinformatics pipeline (https://github.com/EDitt/Sunflower_RNAseq) as implemented in [41]. Raw sequence reads were processed by removing reads containing adapter sequences, as well as unknown or low-quality bases, using Trimmomatic v0.36 [126] with its default settings. Cleaned reads were aligned to the cultivated sunflower reference genome (XRQv2.0; [127]) using STAR v2.7.9a [128] with default parameters. Next, gene expression abundances were calculated per library using RSEM v1.3.3 [129] with default parameters. Finally, the Bioconductor program edgeR v3.34.0 [130] was used to produce normalized read counts via TMM normalization and to identify differentially

expressed genes (DEGs) across treatments and tissue types. The model used to identify DEGs separated samples by tissue and stress type then compared the expression against control samples of the same tissue type. For a gene to be considered for differential expression analysis, it had to meet a minimum expression threshold of 1 count-per-million in 2 or more libraries within a tissue/treatment group. A total of 39,042 genes were retained for this analysis. A false discovery rate (FDR) of $\leq 0.05$ was used as the threshold for identifying DEGs. At this point, five outlier libraries (P5L, P3R, S1L, C2L, DD3L) were identified based on multidimensional scaling plots and removed from subsequent analyses (S1A and S1B Fig). These corresponded to the five samples with the fewest number of reads in the data set. All DEGs were then classified based on tissue specificity (i.e., leaf-specific, root-specific, or shared) as well as stress specificity (i.e., stress-specific or shared by a particular combination of stresses).

To test whether the number of unique genes found in each stress combination was more or less than expected by chance, we took random samples of genes that passed the minimum expression threshold for each stress 1000 times and then calculated the overlap. The number of random genes sampled for each stress was equal to the number of true DEGs found for that stress. The number of unique genes in each stress combination from each random sample was then used to create a distribution from the 1000 samples for the true results to be tested against. *P*-values were estimated by finding the percentile for which the true value fell in each distribution and multiplying it by 2 for an upper and lower two-tailed test. To further explore the similarity of the transcriptional responses of each tissue type to the various stress scenarios, MDS plots were created from TMM normalized counts for different gene and sample sets using edgeR.

## Gene co-expression network construction

We built a signed gene co-expression network using WGCNA v1.70–3 [131]. All genes used for the differential expression were retained for the network construction, with the exception of 11 genes that did not meet the default variance requirements of WGCNA. Three additional outlier samples were removed based upon distance matrix clustering recommended by WGCNA (S3 Fig). A soft-thresholding power of 14 was used to build a signed network given our number of samples and the lack of scale-free topology association due to variance in the data caused by the treatment design as recommended by the WGCNA manual. We then used the automatic 1-step blockwise network construction approach with a maximum block size of 20,000, splitting the data into two blocks, to generate the topological overlap matrix. We did not determine correlations between the co-expression network and the phenotypes measured as RNA was not extracted from the same plants for which we (destructively) measured phenotypes. Correlations were determined between network modules, treatments, and tissues. To test for the enrichment of genes belonging to specific modules within DEG intersections, we first simulated 1,000 co-expression networks by randomly assigning genes to modules of the same sizes as those in the network built from our experimental data. We then asked which simulated module each DEG belonged to and created a distribution of module membership statistics. Lastly, for each set of DEGs, we asked if there were more or less genes belonging to each module than expected by chance and estimated *P*-values from the distribution of the simulated data.

## Classification, GO, and KEGG term enrichment of DEGs

A list of gene annotations and a gene ontology (GO) index were downloaded for the XRQv2.0 reference genome (http://www.heliagene.org/). A GO enrichment analysis was then performed for all DEGs in each tissue, stress combination, and network module using GOseq v.1.44.0

[132] to normalize for gene length bias. A significance threshold of P < 0.05 was used to determine the significance of GO term enrichment after a multiple hypothesis correction using the Benjamini-Hochberg methodology [124]. KEGG terms were also mapped to genes in the XRQv2 genome using BlastKOALA [133] and analyzed for enrichment using the same approach as the GO term enrichment analysis. The results were then mapped to metabolic pathways using the KEGG Mapper—Search tool with the Reference option selected [134].

## Supporting information

**S1 Fig. Multidimensional scaling (MDS) plots created from various sample and gene sets to show how samples differentiate across gene expression space.** Samples are labeled using a common naming theme: The first letters correspond to the stress treatment (DD = dry-down, P = PEG, S = salt, N = low-nutrient), followed by a number representing the identity of that sample, followed by another letter corresponding to tissue type (L = leaf, R = root). The set of samples and genes used for each MDS plot are described in the label for each subfigure. (PDF)

**S2 Fig. Visual depiction of correlations between gene co-expression modules and tissue/ treatment with the inclusion of control samples.** Correlation values (upper text) and *P*-values (lower, parenthetical text) are presented in each cell. Color is determined by the sign and magnitude of the correlation. Positive correlations (red) indicate genes within a module are upregulated within a stress/tissue combination while negative correlations (blue) indicate that genes within the module are downregulated. (TIF)

**S3 Fig. Dendrogram of sample clustering created prior to the construction of the gene co-expression network for the purpose of identifying and removing outliers.** (TIF)

**S1 Table. Table of differentially expressed genes and their expression data.** Each tab is a different test of differential comparison. The leaf_DEGs and root_DEGs tabs test all four stresses against the control treatments for leaf and root tissue respectively. The following tabs are identified using a common naming theme: The first letters in the tab correspond to the stress being compared against the control treatments (DD = dry-down, P = PEG, S = salt, N = low-nutrient) and the next letter corresponds to the tissue type (L = leaf, R = root) followed by "_DEGs". (XLSX)

**S2 Table. Table listing all the genes belonging to each intersection displayed in Fig 2.** Each tab contains a list of gene identifiers and the co-expression network module that gene belongs to. Tabs are named using one letter abbreviations to denote each stress (D = dry-down, P = PEG, S = salt, N = low-nutrient) with multiple stresses in combination denoted using more than one abbreviation followed by an underscore and the tissue type ("_Leaf" or "_Root"). (XLSX)

**S3 Table. Table listing the output of GO term enrichment analysis for each set of DEGs and each intersection of DEGs displayed in Fig 2.** Each tab contains a GO term along with over represented and under represented P-values, the number of DEGs found belonging to that category, the total number of genes belonging to that category, the description of the GO term, and the GO ontology category that term belongs to. Each tab is labeled by stress (D = dry-down, P = PEG, S = salt, N = low-nutrient), tissue type (L = leaf, R = root), and a

descriptor of the set of genes tested for enrichment (allDEGs = the set of all DEGs from that stress, Unique = the DEGs that were unique to a given stress, Intersect = the set of DEGs that are shared among the stresses listed in the tab name).
(XLSX)

**S4 Table. Table listing the output of KEGG term enrichment analysis for each set of DEGs and each intersection of DEGs displayed in Fig 2.** Each tab contains a KEGG term along with the description of the KEGG term, over represented and under represented P-values, the number of DEGs found belonging to that category, and the total number of genes belonging to that category. Each tab is labeled by stress (D = dry-down, P = PEG, S = salt, N = low-nutrient), tissue type (L = leaf, R = root), and a descriptor of the set of genes tested for enrichment (allDEGs = the set of all DEGs from that stress, Unique = the DEGs that were unique to a given stress, Intersect = the set of DEGs that are shared among the stresses listed in the tab name).
(XLSX)

**S5 Table. Table listing the output of GO term enrichment analysis for each co-expression network module.** Each tab contains a GO term along with over represented and under represented P-values, the number of DEGs found belonging to that category, the total number of genes belonging to that category, the description of the GO term, and the GO ontology category that term belongs to. Each tab corresponds to a different module in the network.
(XLSX)

**S6 Table. Table listing the output of KEGG term enrichment analysis for each co-expression network module.** Each tab contains a KEGG term along with the description of the KEGG term, over represented and under represented P-values, the number of DEGs found belonging to that category, and the total number of genes belonging to that category. Each tab corresponds to a different module in the network.
(XLSX)

**S7 Table. Table displaying the results of the module/intersection enrichment analysis.** Results are broken down into two tabs, one for leaf tissue and one for root tissue. The first column represents each stress and stress combination through single letter abbreviations (D = dry-down, P = PEG, S = salt, N = low-nutrient), the second column lists the module being tested for enrichment within the given intersection, the third column lists the under represented P-value, and the fourth column represents the over represented P-value.
(XLSX)

## Acknowledgments

We thank the members of the Burke lab as well as Lisa Donovan, Andries Temme, Loren Rieseberg, Jim Leebens-Mack, Katrien Devos, and Scott Jackson for providing project and manuscript feedback. We also thank Nicole Torralba for her role in plant maintenance and phenotyping, as well as Karolina Heyduk, Jeremy Ray, and Emily Dittmar for their assistance with the transcriptomic analyses. Sequencing services were provided by the Georgia Genomics and Bioinformatics Core.

## Author Contributions

**Conceptualization:** Rishi R. Masalia, John M. Burke.

**Formal analysis:** Max H. Barnhart, Rishi R. Masalia.

**Methodology:** Rishi R. Masalia, Liana J. Mosley.

**Project administration:** John M. Burke.

**Visualization:** Max H. Barnhart.

**Writing – original draft:** Max H. Barnhart, Rishi R. Masalia.

**Writing – review & editing:** Max H. Barnhart, John M. Burke.

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
