## [Decision Letter · Decision Letter 0]

29 Jun 2022

PONE-D-22-01728

Phenotypic and transcriptomic responses of cultivated sunflower seedlings (Helianthus annuus L.) to four abiotic stresses

PLOS ONE

Dear Dr. Barnhart,

Thank you for submitting your manuscript to PLOS ONE. After careful consideration, we feel that it has merit but does not fully meet PLOS ONE’s publication criteria as it currently stands. Therefore, we invite you to submit a revised version of the manuscript that addresses the points raised during the review process.

We look forward to receiving your revised manuscript.

Kind regards,

Mayank Gururani

Academic Editor

PLOS ONE

“We thank the members of the Burke lab as well as Lisa Donovan, Andries Temme, Loren Rieseberg, Jim Leebens-Mack, Katrien Devos, and Scott Jackson for providing project and manuscript feedback. We also thank Nicole Torralba for her role in plant maintenance and phenotyping, as well as Karolina Heyduk, Jeremy Ray, and Emily Dittmar for their assistance with the transcriptomic analyses. Sequencing services were provided by the Georgia Genomics and Bioinformatics Core. This work was supported by a grant to JMB from the NSF Plant Genome Research Program (IOS-1444522).”

“This work was supported by a grant from the NSF Plant Genome Research Program (IOS-1444522; https://beta.nsf.gov/funding/opportunities/plant-genome-research-program-pgrp) to JMB. The funders had no role in study design, data collection and analysis, decision to publish, or preparation of the manuscript.”

Reviewers' comments:

Reviewer's Responses to Questions

**Comments to the Author**

1. Is the manuscript technically sound, and do the data support the conclusions?

Reviewer #1: Yes

Reviewer #2: Yes

Reviewer #3: Partly

2. Has the statistical analysis been performed appropriately and rigorously? 

Reviewer #1: Yes

Reviewer #2: Yes

Reviewer #3: Yes

3. Have the authors made all data underlying the findings in their manuscript fully available?

Reviewer #1: Yes

Reviewer #2: Yes

Reviewer #3: Yes

4. Is the manuscript presented in an intelligible fashion and written in standard English?

Reviewer #1: Yes

Reviewer #2: Yes

Reviewer #3: Yes

5. Review Comments to the Author

Reviewer #1: The various abiotic stresses and the not sufficient nutrient supply result in a great yield loss. The authors compared the effect of four stresses on different phenotypic traits and the transcriptome of sunflower. The topic is interesting, since usually the effect of one stress is only studied. The introduction introduces the scientific background of the research work well. The description of the methods is appropriate. The description of the results is adequate. Some parts of the discussion are the repetitions of the results; therefore, they can be deleted. It would be interesting to determine some regulatory or metabolic pathways based on the gene expression data, which could be responsible for certain observed phenotypic changes. It would be useful to validate the transcriptome analysis by qRT-PCR in the case of selected genes and make a correlation analysis between the expression data of the two methods.

Reviewer #2: Line 161 Table 1: Phenotypic means and standard deviations of all measured traits (n = 21)

Leaf [chlorophyll] PEG= 21.54 ± 0.61b

Salt= 25.13 ± 0.51c

Why the values are higher in such stresses in comparison with control. The chlorophyll content should also be well correlated with a spectrophotometric estimation method apart from a sensor based physical method.

Reviewer #3: The authors have generated data on phenotypic and transcriptomics using sunflower seedlings under different stresses. The seedlings were grown under dry down, PEG and salinity. These stresses were compared with low nutrient stress for identifying commonality genes through transcriptome studies. The authors worked and generated tremendous data. However, there is no novelty in the work. Hence, It is not recommended for the publication.

6. PLOS authors have the option to publish the peer review history of their article (what does this mean?). If published, this will include your full peer review and any attached files.

Reviewer #1: No

Reviewer #2: No

Reviewer #3: No

---

## [Author Response · Author response to Decision Letter 0]

5 Sep 2022

Dear Editor Mayank Gururani,

We thank you for your consideration of our manuscript titled Phenotypic and transcriptomic responses of cultivated sunflower seedlings (Helianthus annuus L.) to four abiotic stresses. We have addressed the comments from editors and reviewers and are submitting a revised version of this manuscript for your consideration. In brief, we have made modifications to ensure our manuscript fits the publication requirements of PLOS ONE, reduced redundancy present in the results and discussion, and better explained certain aspects of our results. Please see our response to the individual reviewer comments following this letter. Our response to the reviewers will be written in bold font.

Sincerely,

Max Barnhart

Changes to Funding Statement

We have removed funding information from our acknowledgements section. The current version of the Funding Statement is accurate.

Response to Reviewers

Reviewer #1: The various abiotic stresses and the not sufficient nutrient supply result in a great yield loss. The authors compared the effect of four stresses on different phenotypic traits and the transcriptome of sunflower. The topic is interesting, since usually the effect of one stress is only studied. The introduction introduces the scientific background of the research work well. The description of the methods is appropriate. The description of the results is adequate. Some parts of the discussion are the repetitions of the results; therefore, they can be deleted. It would be interesting to determine some regulatory or metabolic pathways based on the gene expression data, which could be responsible for certain observed phenotypic changes. It would be useful to validate the transcriptome analysis by qRT-PCR in the case of selected genes and make a correlation analysis between the expression data of the two methods.

When transcriptomic analyses were still being done via array hybridization, qRT-PCR was a standard expectation due to reproducibility issues. With RNAseq data, this step is generally regarded as unnecessary. We do see potential value in using qRT-PCR to extend the results for specific genes to additional samples, but that is beyond the scope of the present paper. 

We have conducted a KEGG term analysis to map gene expression data and network modules to metabolic pathways. To summarize, we found a large number of enriched KEGG terms in the set of root DEGs that belonged to a ‘Biosynthesis of secondary metabolites’ pathway, however there was no clear indication of a shared response between stresses belonging to a highly specific metabolic pathway within sets of DEGs or the network modules. We have added discussion of these results in lines 332-341, 392-397, 496-499, and 761-763 of the track changes manuscript and provided the data in supplementary tables S4 and S6.

We looked for redundancies from the results in the discussion section and found few. In the remaining instances where we have restated things, we found it necessary to do so in order to best explain our interpretation of the data.

Reviewer #2: Line 161 Table 1: Phenotypic means and standard deviations of all measured traits (n = 21)

Leaf [chlorophyll] PEG= 21.54 ± 0.61b

Salt= 25.13 ± 0.51c

Why the values are higher in such stresses in comparison with control. The chlorophyll content should also be well correlated with a spectrophotometric estimation method apart from a sensor based physical method.

In lines 474-475 of the Discussion in the track changes manuscript, we described how the observed increase in chlorophyll content for seedlings under PEG and salt stress is likely due to an increase in LMA that was also observed for those stresses and provided a relevant citation. Increased LMA is due to a thicker leaf, therefore more chlorophyll is detected by the sensor due to the increased mass and volume of leaf in the area in which the measurements are taken. This is not necessarily indicative of more chlorophyll content per unit mass of leaf. We have reworded the passage referenced above to clarify. Given that this result aligns with expectations in the literature for the reason mentioned above, we did not pursue alternate methods of measurement.

Reviewer #3: The authors have generated data on phenotypic and transcriptomics using sunflower seedlings under different stresses. The seedlings were grown under dry down, PEG and salinity. These stresses were compared with low nutrient stress for identifying commonality genes through transcriptome studies. The authors worked and generated tremendous data. However, there is no novelty in the work. Hence, It is not recommended for the publication.

The PLOS ONE website states that: “We evaluate submitted manuscripts on the basis of methodological rigor and high ethical standards, regardless of perceived novelty.” We also note that Rev1 commented specifically on the topic being interesting, particularly given the inclusion of and comparison across multiple stresses while many such studies focus on just one such stress. Regardless, in accordance with journal guidelines, we believe that a perceived lack of novelty should not impact the decision to publish our manuscript.

Abstract: Too lengthy which was not summarized as per the journal concerns.

We have revised our abstract to be more succinct.

61. Climate change is expecting frequent and longer droughts….How this current work tackle the drought stress in plants.

Our work evaluates and compares two water limitation stresses, a dry-down stress and an osmotic stress implemented with PEG. Both of those stresses reduce water availability to plants and simulate a drought stress. Both techniques have also been used to simulate drought stress in other published studies, many of which are cited in our manuscript. Notably, however, we found that osmotic stress is not a good proxy for true water limitation. We believe that this is an important finding that should be shared as it will help to guide future work in this area.

94. Arrange citations in year wise

We have revised our citations and references to adhere to journal style.

106. Since many years scientists are claiming the different genes are involving in signaling pathways. Does really help these pathways to protect the plants particularly abiotic stresses?

The experiment described in this manuscript is not designed in such a way where we can evaluate how gene expression might ‘protect’ a plant from abiotic stress, and we have been careful to avoid making such claims. We only sought to characterize the phenotypic and transcriptomic responses to these stresses and their overlap. These specific goals are stated in the final sentence of the Background (lines 152-157 of the track changes manuscript). While such work does have the potential to highlight candidates for genes of interest in the context of stress tolerance, it does not provide rigorous functional insights. We are thus not comfortable making further assertions regarding the role of these genes in conferring stress tolerance. Further functional characterization, while valuable, is beyond the scope of the present manuscript.

There is no proper conclusion in the manuscript.

The submitted version of our manuscript included a conclusions section that summarized the most important points of our manuscript. Without more specific guidance as to how this section should be modified or improved, we have retained that section as-is.

---

## [Decision Letter · Decision Letter 1]

19 Sep 2022

Phenotypic and transcriptomic responses of cultivated sunflower seedlings (Helianthus annuus L.) to four abiotic stresses

PONE-D-22-01728R1

Dear Dr. Barnhart,

We’re pleased to inform you that your manuscript has been judged scientifically suitable for publication and will be formally accepted for publication once it meets all outstanding technical requirements.

Kind regards,

Mayank Gururani

Academic Editor

PLOS ONE

Additional Editor Comments (optional):

Reviewers' comments:

Reviewer's Responses to Questions

**Comments to the Author**

1. If the authors have adequately addressed your comments raised in a previous round of review and you feel that this manuscript is now acceptable for publication, you may indicate that here to bypass the “Comments to the Author” section, enter your conflict of interest statement in the “Confidential to Editor” section, and submit your "Accept" recommendation.

Reviewer #1: All comments have been addressed

Reviewer #2: All comments have been addressed

2. Is the manuscript technically sound, and do the data support the conclusions?

Reviewer #1: Yes

Reviewer #2: Yes

3. Has the statistical analysis been performed appropriately and rigorously? 

Reviewer #1: Yes

Reviewer #2: Yes

4. Have the authors made all data underlying the findings in their manuscript fully available?

Reviewer #1: Yes

Reviewer #2: Yes

5. Is the manuscript presented in an intelligible fashion and written in standard English?

Reviewer #1: Yes

Reviewer #2: Yes

6. Review Comments to the Author

Reviewer #1: (No Response)

Reviewer #2: All necessary changes suggested have been well addressed. The questions have been answered and that all responses meet formatting specifications.

7. PLOS authors have the option to publish the peer review history of their article (what does this mean?). If published, this will include your full peer review and any attached files.

Reviewer #1: No

Reviewer #2: No

---

## [Editor Report · Acceptance letter]

22 Sep 2022

PONE-D-22-01728R1 

Phenotypic and transcriptomic responses of cultivated sunflower seedlings (*Helianthus annuus* L.) to four abiotic stresses. 

Dear Dr. Barnhart:

I'm pleased to inform you that your manuscript has been deemed suitable for publication in PLOS ONE. Congratulations! Your manuscript is now with our production department. 

Kind regards, 

on behalf of

Dr. Mayank Gururani 

Academic Editor

PLOS ONE